# The journey through care: study protocol for a longitudinal qualitative interview study to investigate the healthcare experiences and preferences of children and young people with life-limiting and life-threatening conditions and their families in the West Midlands, UK

Sarah Mitchell,[1] Anne-Marie Slowther,[1] Jane Coad,[2] Jeremy Dale[1]

[1]Warwick Medical School, University of Warwick, Coventry, UK
[2]Faculty of Health and Life Sciences, Coventry University, Coventry, UK

**Correspondence to**
Dr Sarah Mitchell;
sarah.j.mitchell@warwick.ac.uk

## ABSTRACT

**Introduction** The number of children and young people living with life-limiting and life-threatening conditions is rising. Providing high-quality, responsive healthcare for them and for their families presents a significant challenge. Their conditions are often complex and highly unpredictable. Palliative care is advocated for people with life-limiting and life-threatening conditions, but these services for children are highly variable in terms of availability and scope. Little is known about the lived experiences and preferences of children and their families in terms of the palliative care that they do, or do not, receive. This study aims to produce an in-depth insight into the experiences and preferences of such children and families in order to develop recommendations for the future provision of services. The study will be carried out in the West Midlands, UK.

**Methods and analysis** A qualitative study comprising longitudinal interviews over a 12-month period with children (aged 5–18 years) living with life-limiting or life-threatening conditions and their family members. Data analysis will start with thematic analysis, followed by narrative and cross-case analysis to examine changing experiences and preferences over time, at the family level and within the wider healthcare system. Patient and public involvement (PPI) has informed the design and conduct of the study. Findings will be used to develop recommendations for an integrated model of palliative care for children in partnership with the patient and public involvement (PPI) group.

**Ethics and dissemination** Ethical approval was granted in September 2016 by the National Health Service Health Research Authority (IRAS ID: 196816, REC reference: 16/WM/0272). Findings will be of immediate relevance to healthcare providers, policy-makers, commissioners and voluntary sector organisations in the UK and internationally. Reports will be prepared for these audiences, as well as for children and their families, alongside academic outputs.

## Strengths and limitations of this study

► An in-depth, contextual, longitudinal qualitative study with multiple child and family member stories captured over time.

► New insights will be provided because all of the children and families included in the study could benefit from palliative care as it is currently defined, however, not all will have had conversations about this or have been referred to specialist palliative care services. Findings will focus on healthcare, but there is a wider applicability and relevance to social care and joint planning of services.

► A diverse study population in terms of age, clinical condition, cultural background and family structure will allow detailed consideration of the role of healthcare services in effectively recognising and supporting children and families with their individual needs. However, all will speak English.

► Neonates, preschool children and young people at transition (over the age of 18 years) are all excluded and warrant research in their own right.

► There are multiple potential sources of bias which will be addressed throughout the study, including recruitment bias and the unconscious bias of the researcher.

## INTRODUCTION

Children and young people with life-limiting conditions and life-threatening conditions represent a growing concern in health-care.[1] With advances in clinical practice, the number of children living with these conditions is rising.[1–3] The nature of their conditions is complex and unpredictable; the risk of a sudden deterioration and death is an

---

**Box   Children's palliative care definitions[15 62]**

Palliative care for children with life-threatening conditions is defined by WHO as 'a special, albeit closely related field to adult palliative care; the principles apply to other paediatric chronic disorders:
► The active total care of the child's body, mind and spirit, and support to the family.
► It begins when illness is diagnosed, and continues regardless of whether or not a child receives treatment directed at the disease.
► Health providers must evaluate and alleviate a child's physical, psychological and social distress.
► Effective palliative care requires a broad multidisciplinary approach that includes the family and makes use of available community resources; it can be successfully implemented even if resources are limited.
► It can be provided in tertiary care facilities, in community health centres and even in children's homes'.
The UK national charity for paediatric palliative care, Together for Short Lives, defines palliative care for children with life-limiting conditions as 'an active and total approach to care, from the point of diagnosis or recognition, embracing physical, emotional, social and spiritual elements through to death and beyond. It focuses on enhancement of quality of life for the child and support for the family and includes the management of distressing symptoms, provision of short breaks and care through death and bereavement'.

everyday reality for many. Family carers can experience enormous emotional, physical and financial pressures.[4]

Research suggests that families of children with life-limiting and life-threatening conditions wish for continuous and holistic healthcare, with the option that this is delivered in the home environment.[4] However, most children who die have a pre-existing life-limiting condition,[5] and most die in hospital,[6 7] most frequently in an intensive care environment, where the mode of death is often withdrawal or limitation of life-sustaining treatments.[8–10] The length of stay in the intensive care unit before death is increasing and the costs of hospital care at the end of life are significant.[11 12] This situation presents complex clinical and ethical dilemmas at individual, organisational and societal levels.

Palliative care is an approach to care, which is advocated for all people who live with a life-limiting or life-threatening condition. Current definitions of palliative care are broad (box), which can cause difficulties and lack of clarity for those designing and commissioning specific services.

For clarity, in this paper, children and young people will be referred to throughout as 'children'.

## Which children?
There has been a range of previous work concerned with the identification of clinical conditions where palliative care could be beneficial,[13 14] and the following categorisation is provided by Together for Short Lives[15]:
► *Group 1:* life-threatening conditions where access to palliative care services is necessary alongside attempts at curative treatment and/or if treatment fails, such as cancer.

► *Group 2:* conditions such as Duchenne muscular dystrophy, where premature death is inevitable, but where there may be long periods where the child is well.
► *Group 3:* progressive conditions without curative treatment options, such as Batten disease.
► *Group 4:* irreversible but non-progressive conditions, with complex disabilities and healthcare needs which lead to increased likelihood of premature death, such as severe brain injury.

## Organisational issues
Currently, there is a wide geographic variation in terms of paediatric palliative care services, and a poorly understood range of commissioning arrangements to support these services. Many services exist as a result of significant contributions from the voluntary sector (including children's hospices), through the efforts of motivated individuals, and through non-recurring funding opportunities rather than the implementation of national policy.[16] A significant development is the emergence of paediatric palliative medicine as a subspecialty of paediatrics.[17 18]

Effective palliative care services for children require strong partnerships between providers, and may require cross-boundary, collaborative commissioning between the statutory and voluntary sectors.[19] In the UK, palliative care for children has specifically been included in national policy, a service specification for paediatric palliative care exists and NICE (National Institute for Health and Care Excellence) Guidelines for end-of-life care for infants, children and young people were published in 2016.[20–24]

## Ontology, epistemology and theoretical perspective
Much of the evidence base that guides policy and practice in medicine is derived from experimental research grounded in a positivist paradigm, for example, randomised controlled trials, where a hypothesis can be generated and tested. The positivist approach does not lend itself to research which aims to investigate more complex interventions, such as palliative care, and an interpretive approach is more appropriate. The experience of healthcare services by children with a life-limiting or life-threatening condition and members of their family are shaped and influenced by many interlinked factors including their own personal experiences, values and cultural influences, the values of the healthcare team and the healthcare system, the specific context in which care is delivered and the relationships between those involved in providing and receiving care.[25–27]

The proposed research seeks to understand the mechanisms and influences that shape the experience of care in order to inform both the development and implementation of policy for palliative care services for young children with life-limiting conditions. The methodological approach identified as most appropriate for this research aim is realism, an approach which is increasingly used in healthcare and health sector management research.[28–31]

First described by Bhaskar in the 1970s,[32] and subsequently by Pawson and Tilley [33] realism seeks to understand how phenomena come about as a result of hidden mechanisms, enacted under certain circumstances.[33 34] It acknowledges that there are a wide variety of dynamic contexts and mechanisms which can affect outcomes, including geographical and environmental factors, social and cultural issues and historical factors, and provides a generative approach allowing for the proposal of theories to guide the implementation of policy into practice.[35]

### Rationale for research

Despite the range of recommendations for the provision and development of paediatric palliative care services, there remains a lack of research evidence to support the implementation of these guidelines.[1 18 36–38] The proposed research seeks to address this gap using a realist approach to address research questions that correlate with the practical concerns associated with service delivery. The findings and theories that are generated will provide in-depth insights that will be of immediate relevance to clinicians, commissioners and policy makers, as well as to patients and their families.

### Research questions

1. How do children with life-limiting and life-threatening conditions and their family members perceive healthcare services, and in particular 'palliative care'?
2. What are the experiences and preferences of children living with a life-limiting or life-threatening condition and/or their families, in relation to the delivery of healthcare services?
3. What are the facilitators and barriers to the delivery of palliative care for children, and how might these be overcome?
4. What should an integrated model of palliative care for children look like?

## METHODS AND ANALYSIS

In order to conduct an exploration of the experiences of healthcare from the perspective of children with life-limiting and life-threatening conditions and their families, we will adopt qualitative research methods and a narrative-based approach, suitable for complex, emotionally charged subject areas.[39 40] Active listening, reflection, a flexible approach and insight into the narratives being co-constructed between participant and researcher will be necessary throughout.

This is the protocol for an in-depth longitudinal qualitative study using semistructured interviews with school-aged children (5–18 years) and one or two of their household family members.

Benefits of longitudinal studies include being able to describe the changing needs of the children and their families, and their experience of services, over time,[41] and enabling rapport to build between researcher and participant.

Neonates, preschool children and young people aged over the age of 18 years are excluded from this study. Specific issues around healthcare services arise when considering neonatal care and young people who are making the transition from paediatric to adult services, both of which warrant research in their own right. Research methods would need to be tailored to interview preschool children; this is also an area for potential future research.

The research plan has been informed by review of relevant literature, patient and public involvement (PPI) work and advice from local experts via the West Midlands Paediatric Palliative Care Network.

### Sampling and recruitment

Recruitment to a study of this nature depends on many factors, including the clinical condition of the child, conflicting demands on the family's time and the motivation and understanding of their clinical teams. Recruitment began following ethical approval in October 2016 and will continue until January 2018.

The approach to participants is through:
1. direct invitation via their clinical team
2. leaflets and posters displayed in public areas in the hospital (such as notice boards on wards and in outpatients).

The research will be introduced to clinical teams in both the hospital and the community through formal presentations at departmental meetings and to individual clinicians at their request, as well as to the paediatric palliative care network. The researcher, SM, will undertake a period of shadowing with clinical teams, on hospital wards, in outpatient clinics and in the community.

Potential participants will be provided with a participant information sheet, with details of the researcher, the project, how to get involved and a contact telephone number and email address.

Inclusion and exclusion criteria are outlined in table 1. Our aim initially is to purposively sample children so that each of the four Together for Short Lives categories are represented. However, since children live with such individual and highly complex conditions, we anticipate that achieving this may be difficult. The study population will therefore be children with life-limiting or life-threatening conditions, aged from 5 to 18 years, and their family members, some of whom have experience of a palliative care service, and some who do not.

The study has been carefully designed to ensure that all of the children have the opportunity to participate and that wherever possible the views of the child are included, by tailoring each individual interview to their needs and capabilities (including the consent and agreement process). This may include having a learning disability or communication difficulties associated with their condition.

Ethical approval has been granted for the recruitment of 12–14 families to take part in a series of interviews (longitudinal interviews). The aim is to continue

| Table 1 | Inclusion and exclusion criteria |
|---|---|
| Inclusion criteria | 1. Children aged 5–18 years (school age) with a life-limiting or life-threatening condition who are under the care of the community children's nursing team and/or the children's hospital and who either:<br>► receive palliative care services<br>► are aware of (have had discussions about) palliative care services<br>► are living with relapsing or refractory disease<br>► or have had a life-threatening episode (admission to the paediatric intensive care unit). |
| Exclusion criteria | 2. Their family members, who live in the same household.[63]<br>► Children aged <5 years and >18 years.<br>► Families of children <5 years and >18 years old.<br>► Children and families with whom the research team has clinical contact.<br>► Children and/or families who do not wish to participate.<br>► Children who are too unwell will not be approached for interview, but their family members may still participate if they wish to. The researcher will take advice from parents about when an individual child is 'too unwell' to take part.<br>► Children who are unable to participate in a conversational interview for any reason related to their condition will not be approached for interview, but their family members may participate if they wish to.<br>► Children and families who are unable to provide informed consent in English will not be approached for interview. |

to conduct interviews until data saturation is achieved,[42] however, given the uniqueness and individuality of the stories of children and families, it is possible that new themes will continue to emerge such that data saturation is impossible. We will aim for saturation of the main themes that emerge from the data, and identify emergent themes, which may form the basis of future research.

### Interview plan

Interviews will be carried out by SM, a researcher who is also a general practitioner (GP) with advanced communication skills training and previous experience in qualitative interviewing. According to the preference of participants, interviews will be conducted with individuals, or with the child and parent together, in their preferred location. One or two family members will be interviewed in each family, either individually or together, depending on their preference and what is most convenient for them.

Interviews will be open and conversational, using a blended approach of interview techniques, with passive interviewing allowing the participant space and time to tell their story (narrative), and more active techniques, including appreciative inquiry, which asks 'What works well?' and 'Why does it work well?',[4 43] employed. A topic guide (table 2) will guide the interview; this will continue to develop iteratively throughout the research, with adaptations made during each interview and in response to each individual participant.

For interviews with children, a range of techniques will be used including depersonalising questions, developing a narrative in the third person, and using props and toys to encourage storytelling. Arts-based activities will be used, where appropriate, as a mutual point of focus for the researcher and participant, or as a focus of the interview, as in the draw–write–tell technique.[44] PPI advice

has been sought on the format of interviews for children (table 3).

Each interview will be audiorecorded, with field notes made to include any additional comments from the child or family made once the audiorecording has stopped,[45] reflections on the interview and observation of the family situation, environment, behaviour and any other interactions that may take place (for example, with other members of the family and clinical staff in hospital or on the phone).

Participants will be asked whether they would like to participate in interviews that will take place over a period of up to 12 months. These are intended to allow the identification of common themes over time and for theories generated through analysis of earlier interviews to be tested out during later interviews.[41] The time intervals between interviews will be individually agreed, depending on the child and family circumstances. The method of communication with each family will also be individually agreed (phone or email). Up to three interviews with each participant are aimed for.

We anticipate practical challenges with conducting longitudinal interviews relating to fluctuations and changes in the clinical condition of each child. Depending on their condition, some children will respond well to treatments and get better. Others may suffer unexpected complications of their condition or treatment, and some may suffer deteriorations which bring about the possibility of dying. To manage the research in this context, we will check the family understanding of the situation before every interview. On occasions, interviews may need to be postponed and rearranged at late notice due to a change in circumstances.

For children who are unable to participate in interviews due to their condition, family members will be

**Table 2** Topic guide

| For all families | For those aware of 'palliative care' |
|---|---|
| **Introduction** | **Palliative care and you (if appropriate)** |
| **Please tell me your story, in any way that you can/want to** | ► Do you have 'palliative care' services? |
| ► Please tell me the story of you | ► Have you ever heard the term 'palliative care'? |
| ► Please can you tell me about you? | ► What does that mean to you? |
| ► Your family? | **What do you receive those services for?** |
| ► Your child(ren) | ► What do these services provide for you? |
| **What is important to you?** | **Does it matter what a service is called?** |
| ► What do you like to do? | **Do you receive services from the hospice?** |
| ► Which places are important to you? | **Can you tell me how you came to receive palliative care/know the palliative care nursing team/the hospice?** |
| ► Where do you spend your time? | ► When were you referred? |
| **Which services are involved in your care?** | ► Who brought it up/made the referral? |
| ► Who comes to see you? | ► How was this discussed with you? |
| ► What do they do? | ► How was that for you/your family? |
| ► What is helpful? | **Do you think that medical/nursing staff receive enough training in this area?** |
| ► What is not? | ► What makes you think that? |
| **Which healthcare professionals do you consider to be key in the delivery of your care?** | **Anything else?** |
| ► What works best? | |
| ► Which services/professionals are most helpful? | |
| ► Which services/professionals do you value most? | |
| ► What does not work? | |
| **How do you think services could be improved?** | |
| **Do you talk to other children/young people/families about your healthcare/services?** | |
| ► What do you tell your friends? | |
| ► What tends to come up in these discussions? | |
| ► Would you recommend these services to others? | |

Questions in bold are leading questions. Bulleted questions are prompts.

interviewed. Children and families are under no obligation to take part in follow-up interviews if they do not wish to. In these cases, and with their consent, data from previous interviews will still be included in the study.

### Data analysis
Interviews will be transcribed verbatim, and NVivo used for data handling. Analysis of interview transcripts and field notes will commence alongside data collection, with an initial broad thematic analysis. All data will be coded, and codes grouped into broad overarching themes.

This initial analysis will be followed by an in-depth, narrative analysis, using structure form analysis to examine not just what is being said, but how it is being said, and to propose what works, for whom, in what circumstance at

a micro (immediate clinical team), meso (local organisation) and macro (wider healthcare system) level perspective.[33 46 47]

The collection of longitudinal data allows for innovative approaches to be taken in data analysis.[41] Matrices will be developed to identify key times for families and identification of cross-cutting themes at these times, for example, the time of diagnosis, an admission to intensive care or referral to a palliative care team.

Peer review and respondent validation will take place throughout the data analysis as follows[48 49]:
1. Peer review: SM will code all of the data. A selection of transcripts will be reviewed and independently coded by other members of the research team in order to

**Table 3** Feedback from PPI groups on interview plans

| January 2016 | "Those who are passionate about improving palliative care will take part regardless of how sensitive this may be" |
|---|---|
| July 2016 | "Remember young people who are seriously ill are more mature, they have to grow up"<br>"Keep it simple as often a child will openly speak anyway"<br>"'Do you talk about it to your friends?' is a good question, a good way to talk to most ages"<br>"Children are more eloquent, mature and more capable than you think" |
| October 2016 | "Use pictures and images, more emojis" |
| February 2017 | "Doesn't make me uncomfortable as I think it is very important and relevant" |

PPI, patient and public involvement.

decrease lone researcher bias.[48] The coding frameworks will be discussed and compared, allowing further development of categories and themes.

2. Respondent validation: by returning to participants to conduct longitudinal interviews, there is an opportunity to check, validate or refute emergent themes from the initial data analysis.

### Healthcare professional perspectives

There are 12 paediatric palliative care networks in the UK, which include professionals from a range of organisations within paediatric palliative care. Several have patient and family representatives. Arrangements will be made to present study findings to four of the UK networks at existing meetings. The presentation will be followed by a structured focus group which will aim to first to test out and validate with palliative care professionals the themes from the research findings, and second aims to collect views of professionals. These multiple perspectives will inform and guide the formulation of recommendations for healthcare services in the future.[50]

An expression of interest email will be circulated to network chairs via Together for Short Lives, and arrangements made to attend meetings from networks who respond. Audiorecorded focus group discussions will be carried out at those meetings by SM.

### Patient and public involvement

PPI has been integral to the design and conduct of the study. Members of existing groups at a children's hospital and children's hospice have provided advice on the study proposal and design. Smaller groups have been recruited for specific activities, including conference presentations. Group members range in age from 12 to 22 years. PPI activities are outlined in table 4, and will continue throughout the project, with the aim of coproducing the recommendations for the model of care. This will involve structured group sessions during which anonymised findings of the data analysis will be presented to the group for feedback and comment. A patient experience framework will be used to structure the discussion and to build recommendations.[51]

### Strengths and limitations

The strength of this study lies in the in-depth, contextual qualitative nature of the data, with multiple child and family member stories captured over time. Our anticipated study population is diverse in terms of age, clinical condition, cultural background and family structure, allowing detailed consideration of the role of healthcare services in effectively recognising and supporting children and families with their individual needs. All of the children and families included in the study could benefit from palliative care as it is currently defined,[15] however, not all will have had conversations about this with their clinicians, or been referred to specialist palliative care services. Given the nature of their clinical conditions, including for some the inability to communicate verbally or deterioration in their health, recruitment and retention within the study is likely to become a challenge and will require a reflexive, flexible approach.

Potential limitations in the study include our exclusion of neonates, preschool children and young people at transition (over the age of 18 years). These groups all warrant research in their own right. Given the time and resource constraints of the study, all interviews will be carried out in English. Further research into the experiences of children and families who cannot communicate in English is necessary. There will be ongoing consideration of sources of bias. Recruitment bias is being addressed by aiming for a diverse sample and providing access to project information independent of the clinical teams. Data saturation will be sought during data analysis, with an ongoing process of reflection and peer review to address any possible unconscious bias of the researcher (SM).

| Table 4 | PPI activities |
|---|---|
| Completed PPI activities | ▶ Developing the original research proposal.<br>▶ Advising on the language used in the study (suggesting a change in the title from 'Palliative Care for Children and Young People: What? When? How?' to 'The Journey through Care'.<br>▶ Developing participant resources including leaflets for older and younger children.<br>▶ Interview design, including suggesting how questions could be phrased and asked.<br>▶ Providing family perspectives to a literature review, and becoming a coauthor on the paper.[18]<br>▶ Taking part in oral presentations at regional conferences. |
| Work in progress | ▶ Designing conference posters and presentations for national conferences.<br>▶ Working as coresearchers to carry out a survey study to investigate understanding of the term 'palliative care' for children and young people and healthcare professionals. |
| Future plans | ▶ Working with a project-specific group to explore the findings of the research study and develop recommendations for a new model of care.<br>▶ Dissemination projects including conference presentations, posters, website design, use of social media, infographics and films. |

PPI, patient and public involvement.

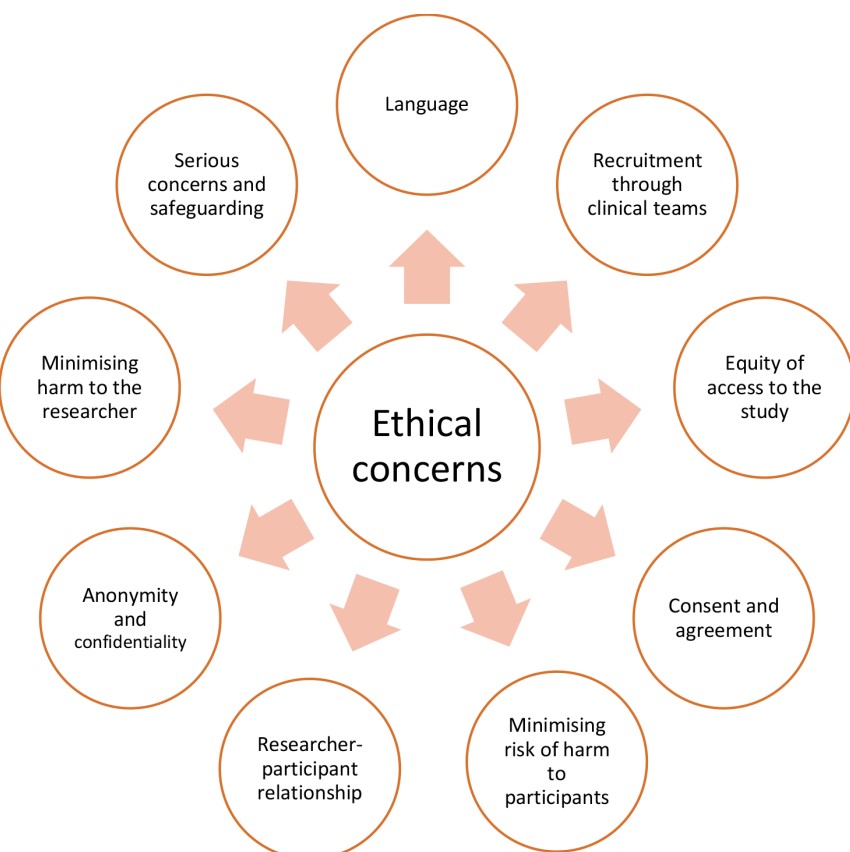

**Figure 1** Ethical issues in longitudinal qualitative research for children and families in palliative care.

## Ethics and dissemination

Research with children raises ethical and legal considerations around recruitment, consent and data collection.[52 53] In addition, research regarding palliative and end-of-life care can be emotionally demanding and distressing for those involved. There are also particular ethical issues to consider given the longitudinal nature of the study.[54]

We are recruiting children and families who are potentially vulnerable and may be experiencing considerable distress. The justification for our approach is that children and their families in this situation are rarely asked about their experiences, but talking to them and understanding their experiences is essential in order to be able to design and develop services that respond to their actual needs. Here, we summarise our approach to the ethical issues the study raises (figure 1).

## Language

Published literature suggests that the term 'palliative care' is poorly understood and perceived negatively,[55–59] a view confirmed by our PPI group. The scope of our study is therefore to investigate the experiences of children with 'life-limiting', 'life-threatening' and 'conditions which may or may not get better', whether or not they have heard of palliative care or receive care from specialist services. 'Palliative care' will be avoided in participant information sheets and interviews, unless individuals are already familiar with palliative care services or bring it up themselves.

## Recruitment

There is an ethical challenge in terms of potential coercion to the study by clinicians who know the family well. Clinicians will therefore only provide study information but will not actively recruit families; the initial expression of interest is from the family to the researcher. The researcher (SM) will then discuss the study in person or by phone with the child and their family member(s) and answer any questions before arranging a time for interview. Participants will be made aware that they can decline to take part or to withdraw at any stage without having to give a reason. Interviews are only carried out at a time that is mutually agreed and minimises any potential inconvenience or intrusion.

## Equity of access to the study

Recruitment through clinical teams is widely used in palliative care research but may be limited by 'gatekeeping'.[60] There may also be families who wish to participate who do not find out about the study through their clinical team. In order to address this, we have designed posters for display on hospital wards and in outpatients, and at the local children's hospice, and a paragraph for organisational newsletters. These provide the direct contact details for the researcher (email, text or phone).

**Table 5** Planned outputs from the research

| Academic/clinical audiences | Patient, public and policy maker audiences |
|---|---|
| ► PhD thesis<br>► Peer-reviewed publications<br>► Presentations at national and international conferences | ► A report prepared for participants and PPI volunteers.<br>► The development of guidance for commissioners and providers.<br>► The development of resources that are accessible to patients and families. |

The study setting is Birmingham, UK, a city where the population is highly diverse in terms of family situation and multiculturalism. Over 50% of families with a child known to palliative care services in Birmingham and Solihull are from black or minority ethnic backgrounds.[61] Many of these families speak English as a first or second language, so within the time and resource constraints of this study, interviews will be carried out with those who can provide informed consent and take part in an interview in English.

## Consent
Consent for the study raises ethical and legal issues with children who are under the age of 16 years and/or do not have the capacity to consent. We will aim for written and/or verbal consent and agreement from every individual for every interview.

For children under the age of 16, written consent will be obtained from the parent and then verbal or written agreement obtained from the child.

In keeping with the Mental Capacity Act, there is an assumption of capacity in young people aged 16 years and over, so they will be asked for consent first, followed by agreement from their parent(s). Parental agreement is not a legal requirement, but conducting an interview with a young person about a potentially difficult subject without the knowledge or agreement of their parents is an ethical concern. If there is a concern that the young person lacks capacity or is considered particularly vulnerable, for example, with a learning disability, parents will be asked to provide verbal and written consent in addition to the young person's agreement.

Parental consent is required for any interview to be carried out in the family home.[53]

For a child on a full care order, social worker consent would replace that of parental consent. Where possible parental consent/agreement will also be sought.

## Interviews
Subject areas discussed during interviews may cause distress to participants, and recruitment may occur soon after sensitive conversations. We have designed the study to ensure that the risks and burden associated with taking part in the study are minimal.

Qualitative interviews will be informal and reflexive to the needs of the participant. In the event that a participant experiences any difficulties during the interview, such as tiredness or distress, the interview will be halted, and if necessary brought to an end. Adequate time will be given for debrief, and the researcher will provide information about local resources for support if necessary. Interviews will be carried out at a time and in a location that is convenient to the participants. If this is in hospital, the researcher (SM) will liaise closely with clinical teams so that the research does not interfere with routine clinical care and ward work.

## Longitudinal interviews
Family views and understanding of what might happen next as a result of the condition of the child will be discussed sensitively, and any follow-up interviews scheduled around possible further treatments. If it seems likely that there will be a deterioration in the condition of the child, this is explored carefully and an agreement made with the individual family about whether they want to continue to participate in the study.

## Anonymity and confidentiality
All interview data will be anonymised with personal identifiers removed. Any qualitative interview data that could identify child, families or any professionals involved in their care because of the individuality and context of the narrative is included in the data analysis, but will be excluded from reporting.

Field notes and anonymised interview transcripts will be stored securely on a password protected university hard drive.

## Minimising harm to the researcher
There is a need for clearly defined boundaries for a researcher–participant relationship in a longitudinal study of this type. It will be made clear to participants at the time of consent that it is not the role of the researcher to provide personal support or clinical advice. With the risk of emotional distress for the researcher, plans to ensure adequate support through regular academic supervision and access to a counsellor are in place.

## Serious concerns and safeguarding
If information contained in a participant's response indicates a serious clinical or safeguarding concern or an issue which may jeopardise the safety of the participant or another person, this will be escalated appropriately in line with the protocols of the community or hospital trusts. This may on very rare occasions necessitate a breach of participant confidentiality in order to maintain their safety. Participants will be informed of any disclosure and to whom it is made.

## Dissemination plan

The research is embedded in plans for impact. Table 5 outlines our planned outputs. We will work on traditional academic and clinical outputs, including manuscripts with the results of the study for publication in a peer-reviewed journal. Simultaneously work will be carried out with the PPI group to plan innovative, accessible outputs for patients, the public and commissioners which will include infographics and film based reports outlining our recommendations.

**Contributors** SM drafted the protocol with regular academic supervision from JD, A-MS and JC. The study was conceptualised by SM, JD, A-MS and JC, informed and guided by patient and public involvement. The protocol incorporates peer review feedback received during the application process for the NIHR Doctoral Research Fellowship, and the MPhil to PhD upgrade panel at the University of Warwick. JD, A-MS and JC have all reviewed the protocol for intellectual content. All authors have reviewed and agreed this version.

**Funding** This work is supported by a National Institute of Health Research Doctoral Research Fellowship (Dr Sarah Mitchell DRF-2014-07-065).

**Disclaimer** This article presents independent research funded in part by the National Institute for Health Research (NIHR). The views expressed are those of the authors and not necessarily those of the NHS, the NIHR or the Department of Health.

**Competing interests** None declared.

**Patient consent** Obtained.

**Ethics approval** Ethical approval was granted by the UK Health Research Authority on 14 September 2016 (IRAS ID: 196816, REC reference: 16/WM/0272, sponsor: University of Warwick).

**Provenance and peer review** Not commissioned; externally peer reviewed.

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
