## [Reviewer comments · BMJ Open]

ARTICLE DETAILS

TITLE (PROVISIONAL)	The Journey through Care: Study protocol for a longitudinal qualitative interview study to investigate the healthcare experiences and preferences of children and young people with life-limiting and life-threatening conditions, and their families in the West Midlands, UK.
AUTHORS	Mitchell, Sarah; Slowther, Anne-Marie; Coad, Jane; Dale, Jeremy

VERSION 1 – REVIEW

REVIEWER	Emma Popejoy University of Nottingham, UK Nottingham Children's Hospital, UK
REVIEW RETURNED	26-Jun-2017

GENERAL COMMENTS	This paper details a study protocol for a hugely relevant and necessary study into children's palliative care services. I think with a few very minor amendments, this paper should be accepted for publication. Suggestions for consideration/revision are as follows: Pg 6 (Methodology) - I am not sure how the positivist approach links with the constructionist philosophy that is discussed. I suspect families will provide different accounts and experiences. If you take a positivist stance that an objective and discernible reality exists, how will you decide/privilege which account accurately represents the objective reality (when potentially conflicting accounts are given)? Also given the interview approach and narrative analysis, I'm not sure how it is possible to be an objective observer (as required in the positivist tradition), rather than being an active co-constructor of the data. Is the study rooted in critical realism instead of positivism? Pg 7 (Sampling and recruitment) In the discussion of the inclusion criteria, you discuss the Together for Short lives categories. Although you discuss these on page 5, you do not refer to them as being TfSL criteria. Perhaps on page 5 you could highlight that these are the TfSL criteria and then refer back to them when you discuss the inclusion criteria on page 7. There appears to be no discussion of how many cases will be recruited, some discussion of a maximum number of cases would be useful, especially given the huge volume of data that will be collected from this longitudinal study.
---

	Pg 8 (Table 1) In the table you state that children won't be interviewed if they are "too unwell" - perhaps you could give some idea of what this means and/or who will decide if the child is "too unwell". You also state that children you have clinical contact with will be excluded. I believe this is the first reference to you having a clinical role, perhaps one sentence to discuss your clinical role and a consideration of why it may be inappropriate to interview your own patients would be useful to include. Pg 12 (Data Analysis) The section on focus groups confused me slightly. It is not clear whether this is to assist in making policy recommendations (in which case I'm not sure it is relevant to be within the data analysis section) or whether the focus groups will assist in the analysis of the data. If it is the latter, I think this may present challenges, given that health professionals/experts have a very different experience to families and may interpret the findings differently. This presents challenges for how you would incorporate their analysis into your own analysis (given that you will be fully immersed in the data and they will not), and may well present some tensions in relation to the underlying philosophy. With a few very minor changes and a consideration by the authors of some of the issues as highlighted above, this paper should be accepted for publication without delay.
--	---

REVIEWER	Lucy Coombes Royal Marsden NHS Foundation Trust and Shooting Star Chase Children's Hospice, UK
REVIEW RETURNED	19-Jul-2017

GENERAL COMMENTS	Abstract:  -The aim at the end of the introduction is a bit ambiguous. The is no information on the population, intervention or setting. -Methods and analysis - the abbreviation CYP is used but is not defined in full anywhere -In the ethics and dissemination - the results will be just as applicable to NHS paediatric palliative care teams. Introduction:  -p6. You may want to consider referencing Richard Hain's work on ICD10 coding and directory of life-limiting conditions in children. Sampling and recruitment:  - p9 - having tried to recruit children with life-limiting conditions to research myself, I imagine you will find it hard to recruit many children and young people. This is because many are non-verbal or not well enough to participate. As you say later on, 50% of the population are from minority groups and I wonder whether you are doing anything specific to recruit here. Otherwise you may find (as many other researchers have) that the population recruit does not reflect the population eligible for the study.
---

	Interview plan: -It is not clear whether you are planning to interview one or both parents? If you interview both I think consideration is needed as to whether they are interviewed together or separately. There is evidence that parents should be interviewed separately as their responses are different if together (Michelson 2009). Table 4: -4th bullet point under completed PPI activities - incomplete sentence Patient and public involvement: -Line 49 and line 54 - the age ranges are not consistent. Strengths and limitations: -I think you need to add something about the challenges of recruiting children as many will be non-verbal/too unwell. Overall, I think this is a really important and interesting study and I wish the researchers every success. I look forward to seeing the results when they are published.
--	--

REVIEWER	Stuart Ekberg Queensland University of Technology, Australia
REVIEW RETURNED	25-Jul-2017

GENERAL COMMENTS	This protocol describes a longitudinal qualitative study designed to explore the healthcare experiences of children and families when a child has a life-limiting or life-threatening condition. This project is unique in several respects; a protocol describing it therefore warrants publication to ensure that the proposed approach can inform ongoing research. I am currently undertaking a systematic review of qualitative research relating to healthcare users' experiences when a child has a life-limiting condition and can confirm that the proposed research has at least three novel aspects:  1. A longitudinal focus: Most existing studies collect data at a single time point, often following the death of a child 2. Inclusion of children: Most studies collect data from families, with only a minority including child patients or their siblings 3. A focus on paediatric palliative care: Although there is a growing body of research considering families experiences of healthcare more generally, there is considerably less work that specifically considers experiences of specialist paediatric palliative care. Although I do not believe that any of the below queries are insurmountable obstacles to publication, it is my opinion that the methodology and method sections deserve more thought and clearer explication. First, I was unclear what was meant by "a 'casual' intellectual puzzle" (p. 6).
---

Second, the authors propose to adopt a positivist approach in their research, and then continue to explain that they will draw upon realism and constructionism. It is not clear why the authors consider constructionism and positivism to be congruent with one another. Consider, for instance, the following:

- “What constructionism drives home unambiguously is that there is no true or valid interpretation. There are useful interpretations, to be sure, and these stand over against interpretations that appear to serve no useful purpose” (Crotty, 1998: 47, emphasis added).

- “Social constructionism insists that we take a critical stance toward our taken-for-granted ways of understanding the world and ourselves. It invites us to be critical of the idea that our observations of the world unproblematically yield its nature to us, to challenge the view that conventional knowledge is based upon objective, unbiased observation of the world. It therefore opposes what is referred to as positivism and empiricism, epistemological positions that are characteristic of the ‘hard’ sciences such as physics and biology” (Burr, 2015: 2, emphasis added).

Third, it is claimed that this study will result in a “meta-synthesis.” This term is usually reserved for the description of systematic reviews of qualitative studies (see, for example, Walsh & Downe, 2005), so it is unclear how a single study has the capacity to achieve this outcome.

In addition to addressing the above queries relating to the methodology section, it is my opinion that the method section would also benefit by clarifying the following:

First, no explicit justification is given for the exclusion of neonates, preschool aged children, and young adults who are more than 18 years old.

Second, the authors could be more explicit about the methods they proposed to use to analyse the data. The abstract mentions that thematic analysis, narrative analysis, and cross-case analysis will be used. However, only narrative analysis is explicitly mentioned in the Data Analysis subsection. Further information is needed in this section, including an explicit description of each approach that will be used and why this is appropriate for the aims of the study.

Third, can the structured group sessions with the PPI group be recorded and included as data in the study? It may be the case that PPI members reflecting on anonymised findings of the data analysis will contribute perspectives from their own experiences that could be used to inform ongoing analysis. It would be a shame to miss out on these.

Finally, I noticed a few minor points that I warrant attention:

- I could not find a point early in the protocol where the reader is informed that the study will be conducted in the UK. Although this progressively becomes apparent, it could be made much clearer.

- At one point in the manuscript the authors claim that they will use ‘children’ to refer to their target population, but then later refer to this group with ‘CYP’ (an abbreviation which is never explicitly defined). A consistent term should be used throughout.

	 The exclusion criteria in Table 1 uses the pronoun 'I'. It is unclear who this refers to. References cited Burr, V. (2015) Social Constructionism (3rd Edition). Hove: Routledge. Crotty M. (1998). The Foundations of Social Research: Meaning and perspective in the research process. Sydney: Allen & Unwin. Walsh, D., & Downe, S. (2005). Meta-synthesis method for qualitative research: A literature review. Journal of Advanced Nursing, 50(2), 204-211.
--	---

VERSION 1 – AUTHOR RESPONSE

I would like to thank the reviewers for their details and considered comments, which were very helpful and thought-provoking. I enclose a revised manuscript with tracked changes, and a clean version, for your consideration for publication in BMJ Open.

We have addressed the incomplete sentence and inconsistency with age ranges. We have removed the abbreviation CYP from the manuscript and used “children” consistently throughout.

In response to the reviewer’s comments, the following changes have been made:

1. The title has been updated to comply with the preferred format of the journal
2. A dissemination plan is outlined.
3. P.2 The aim at the end of the introduction of the abstract has been revised for clarity.
4. P.6 The methodology section has been updated at the end of the Introduction. Ontology and epistemology, methodology and theoretical perspectives are all described in more detail, and the mistake about positivism has been rectified.
5. P.6 it is not a “casual” intellectual puzzle, it is a “causal” intellectual puzzle which hopefully is now clarified with our revisions to this paragraph and in Figure 1.
6. Richard Hain’s ICD-10 work is now included as a reference.
7. We have removed the term “meta-synthesis” to avoid any confusion over this term.
8. P.8 Methods – the age range 5-18 is described. The rationale for this age range is described in more detail in the third paragraph, and is referred to again in Strengths and Limitations.
9. More detail is now provided about the methods of analysis, both through the Methodology section, and in Data Analysis.
10. P. 7-8 the categories are now described as the “Together for Short Lives” categories. The issue of conducting the study in English only is considered later in the article in Strengths and Limitations (p.14).
11. A discussion of the number of cases that will be aimed for has been added (ethical approval has been granted for the recruitment of 12-14 families)
12. Table 1 P.9 More information has been added about when a child would be considered “too unwell” to participate. Details of the researchers clinical background have also been added earlier in the manuscript (in methodology)
13. P.9 Interview plan – revised for clarity to about who will be interviewed, when, how and whether or not interviews will be conducted together or individually (depending on the preference and needs of the family).
14. P.12 the section on focus groups has been retained but amended to provide clarity about the aim of the focus groups.

15. P.13 The reviewer's point about potentially missing valuable perspectives from PPI group members is valid, however we are working to ensure that the patient and public involvement work in this study is a distinct and different part of the research. It is important that it is clearly "involvement" rather than "participation", and therefore the findings from the PPI work will be treated as their own entity, to guide and shape the research.

16. P.14 sentence added to outline the challenge of recruitment of children.

17. Word count is now 3994, with palliative care definitions being presented in Box 1.

Formal ethical approval has been granted by the UK Health Research Authority on 14th September 2016 (IRAS ID: 196816, REC reference 16/WM/0272, Sponsor University of Warwick). The project is funded by the National Institute for Health Research (DRF-2014-07-065), a recognised, open access advocating research-funding body.

The enclosed manuscript has been read and approved by all authors. It is not under active consideration for publication elsewhere, has not been accepted for publication, nor has it been published in full or in part.

VERSION 2 – REVIEW

REVIEWER	Emma Popejoy University of Nottingham, School of Health Sciences and Nottingham University Hospitals NHS Trust
REVIEW RETURNED	21-Aug-2017

GENERAL COMMENTS	I feel that this paper is now ready for publication. The author may wish to consider providing a further explanation of the purpose of the focus groups (on pages 12-13), as this remains unclear to me. However, I do not believe that this is absolutely required for the paper to be accepted, but I personally would be interested and find a further explanation useful. Also, on line 26 (page 2) it says "longitudinal interviews over a ??12 month period" - the question marks just need removing. This looks like a great study and I look forward to reading the results.
---

REVIEWER	Lucy Coombes Royal Marsden NHS Foundation Trust United Kingdom No Competing Interest
REVIEW RETURNED	29-Aug-2017

GENERAL COMMENTS	This is a really important piece of research and I look forward to reading the results. Most of the comments I made initially have been addressed. I understand that R and D and ethics approval has already been granted which is why some of my comments regarding the methodology have not been. Good luck with the research.
--

REVIEWER	Stuart Ekberg Queensland University of Technology, Australia
REVIEW RETURNED	30-Aug-2017

GENERAL COMMENTS	The revisions that have been made have strengthened the manuscript. However, I have a few remaining concerns that I think should be considered prior to publication. I still have reservations about the proposed framework for the study. The authors name a range of concepts – relativism, realism, constructionism, and critical realism – without: a) explaining what each of these mean; and b) outlining the debates that occur within and between these positions. Moreover, the references supplied do not reflect an engagement with the primary proponents of these different positions. These positions are too complex to gloss with a single sentence. I acknowledge the importance of the authors explaining their stance that “reality exists and is actively “constructed” by human beings through actions in a social context, rather than being passively received by them.” However, I think further thought is needed to determine: a) whether all of the named concepts are needed to justify this position; b) the level of detail that is necessary for readers to understand these concepts, why they are appropriate, and any tensions within or between them. The explanation of a “causal” intellectual puzzle is still unclear to me. Is this necessary? If not, you could replace this with something like “frameworks” after “policy and service delivery”. If you did this, Figure 1 would not be necessary. Please note the typographical error (inclusion of two question marks) in the abstract. References cited Crotty, M. (1998) The Foundations of Social Research: Meaning and perspective in the research process. Sydney: Allen & Unwin. Ormston, R., Spencer, L., Barnard, M. & Snape, D. (2013). The foundations of qualitative research. In J. Ritchie, J. Lewis, N.C. McNaughton & R. Ormston (Eds.) Qualitative Research Practice. London: SAGE.
---

VERSION 2 – AUTHOR RESPONSE

Thank-you for the further feedback from the reviewers of this protocol paper. I enclose a revised manuscript with tracked changes, and a clean version, for your consideration for publication in BMJ Open.

Apologies for the typographical error in the abstract which has been corrected.

We have considered the reviewer's comments carefully and have been made the following revisions:

1. A further sentence to clarify the aim of the focus groups has been added.
2. I have revised the section on ontology, epistemology and theoretical perspective to provide more clarity, with focus and justification for a critical realist approach. I hope that this now provides the detail and clarity required for the research protocol. The proposed research will be written up as a PhD thesis, in which the philosophical debates will be explored in much more detail than this protocol.
3. I have removed the "causal" intellectual puzzle sentence and diagram.

Formal ethical approval has been granted by by the UK Health Research Authority on 14th September 2016 (IRAS ID: 196816, REC reference 16/WM/0272, Sponsor University of Warwick).

The project is funded by the National Institute for Health Research (DRF-2014-07-065), a recognised, open access advocating research-funding body.

The enclosed manuscript has been read and approved by all authors. It is not under active consideration for publication elsewhere, has not been accepted for publication, nor has it been published in full or in part.